# Multi-User Visible Light Communication and Positioning System Based on Dual-Domain Multiplexing Scheme

Zhongxu Liu [1] and Changyuan Yu [1,2,*]

1   Department of Electronic and Information Engineering, The Hong Kong Polytechnic University, Kowloon, Hong Kong 999077, China
2   Shenzhen Research Institute, The Hong Kong Polytechnic University, Shenzhen 518000, China
*   Correspondence: changyuan.yu@polyu.edu.hk

**Abstract:** Visible light communication and positioning (VLCP) is a promising candidate for constructing a multi-functional wireless network with large-scale connectivity and centimeter-level positioning. However, there is still a lack of effective methods to offer simultaneous visible light communication (VLC) and visible light positioning (VLP) functions for multiple users. Thus, we propose a multi-user VLCP system based on a dual-domain multiplexing (DDM) scheme, where both the time and code resources are multiplexed to transmit VLCP signals for multiple users simultaneously. In the proposed system, the data of different users are distinguished by using code division multiplexing technology, while the VLCP signals transmitted from different LEDs are separated by adopting time division multiplexing technology. The performances, including bit-error rate and positioning error, are evaluated through both simulation and experimentation to verify the feasibility of the proposed multi-user VLCP system. In the experiment, a VLCP system with four LED transmitters was able to simultaneously support low-speed VLC with free error and accurate VLP with a 2 cm precision for eight users. This offers an effective solution to support a large number of users with simultaneous VLC and VLP functions in the future multi-functional wireless network.

**Keywords:** visible light communication and positioning; visible light communication; visible light positioning; code division multiplexing; time division multiplexing

## 1. Introduction

The sixth-generation (6G) wireless communication network is expected to offer multiple functions, such as ultra-low latency communication, large-scale connectivity, and centimeter-level positioning [1,2]. Recently, due to the advantages of unregulated tremendous spectrum resources, energy conservation, high data confidentiality, and immunity to electromagnetic interference, visible light communication (VLC) has become a promising candidate for 6G communication [3–5]. Meanwhile, visible light positioning (VLP) is another functionality brought about by VLC. Compared to the positioning systems based on radio frequency (RF) technologies, such as WiFi [6], Bluetooth [7], and Zigbee [8], the VLP system can provide a high-precision and low-cost positioning solution, because VLP signals cannot penetrate walls, resulting in less interference and meaning that the existing lighting facilities can be reused, reducing the deployment cost [9,10]. In addition, the communication and positioning functions are the foundation for the emerging Internet of Things (IoT) applications, such as smart manufacturing, sweeping robots, smart medical care, and virtual reality [11]. Therefore, achieving simultaneous multi-user VLC and high-accuracy VLP is crucial to building a multi-functional 6G wireless network.

Extensive studies have been conducted to integrate VLC and VLP and proposed the concept of visible light communication and positioning (VLCP) [12–14]. In VLCP systems, the different user data for VLC and different positioning signals for VLP need to be separated at the receiver to avoid interference, which requires adopting multiplexing technology.

The customarily used multiplexing technology includes time-division multiplexing (TDM), frequency-division multiplexing (FDM), and code-division multiplexing (CDM). In the TDM-based VLCP system, the total time resource is divided into several time slots and assigned for VLC signals and VLP signals, respectively [15,16]. However, this will reduce the transmission efficiency of VLC and also increase the time delay for both VLC and VLP, especially when the VLCP system simultaneously supports a large number of users. In the FDM-based VLCP system, the total frequency resource is divided into two sub-bands for VLC and VLP, respectively [17,18]. However, the VLP signals will occupy the limited bandwidth of commercial LEDs (usually a few MHz), which reduce the capacity of VLC. In addition, the TDM-based solutions and FDM-based solutions require additional guard time slots and bands to avoid interference between VLC and VLP signals, respectively. This reduces the time and spectral utilization. As for the CDM-based VLCP system, a set of spreading codes is divided and assigned to LEDs to distinguish different VLCP signals, then the correlation values between the spreading codes and the VLCP signals are used to recover the user data for VLC and calculate receiver signal strength (RSS) for VLP [19,20]. However, it is hard for the CDM-based solution to support a large number of users because each LED can only support the VLC function for one user, which means the number of LEDs limits the user number. Therefore, new approaches are required to support multiple users for VLCP systems.

In this paper, we propose a multi-user VLCP system based on the dual-domain multiplexing (DDM) scheme, which multiplexes the code and time resources to transmit different VLCP signals for multiple users simultaneously. Specifically, each user is assigned a unique spreading code to encode the user data, and the LED transmitters transmit the VLCP signals in turn. In this way, the data of different users are distinguished by CDM, while the VLCP signals transmitted from different LEDs are separated by TDM. For achieving simultaneous VLC and VLP functions, on the receiver side, the spreading codes are used to decode the received signal to obtain the correlation values. Then, the correlation value can be used to recover the user data for VLC and to calculate the RSS for VLP. We verify the feasibility of the proposed multi-user VLCP system through both simulation and experimentation and investigate the system performances, including bit-error rate (BER) and positioning error. The results show that continuous VLC transmission with accurate VLP function can be provided for multiple users simultaneously. Therefore, the proposed scheme can effectively integrate the VLC and VLP into one system, and the supporting number of users will not be limited by the number of LEDs but can be designed according to the used number of spreading codes.

## 2. Materials and Methods

### 2.1. System Design

The schematic diagram of the proposed multi-user VLCP system is shown in Figure 1a, in which four LEDs are used to provide VLC and VLP functions for N users simultaneously. To simultaneously distinguish the data of different users and the VLCP signals from different LEDs, we adopted the dual-domain multiplexing scheme, which multiplexes the code and time resources. As shown in Figure 1b, a set of spreading codes is assigned to multiple users, while the total time resource is divided into four slots and allocated to four LEDs, respectively. On the transmitter side, the data of each user are encoded by the unique spreading code and accumulated as the transmission data. After on-off keying (OOK) modulation and adding bias current, the transmission data are loaded to each LED, in turn, through TDM. In each time slot, only one LED transmits the VLCP signal to the photodetector (PD) of each user through the VLC channel. On the receiver side, the PD of each user converts the VLCP signal to the electrical received signal. Then, the spreading code is used to decode each bit of the received signal to obtain the corresponding correlation value used for both VLC and VLP. Finally, the correlation value is utilized to recover the user data for VLC through the threshold decision and calculate the received signal strength (RSS) for VLP in each time slot. After all time slots, for each user, three of the

largest RSS are selected to locate the user by using the trilateration algorithm. Therefore, the continuous VLC transmission with VLP function can be realized through the dual-domain multiplexing scheme. In addition, the supporting number of users will not be limited by the number of LEDs but can be designed according to the used number of spreading codes. The principle of the proposed multi-user VLCP system is as follows.

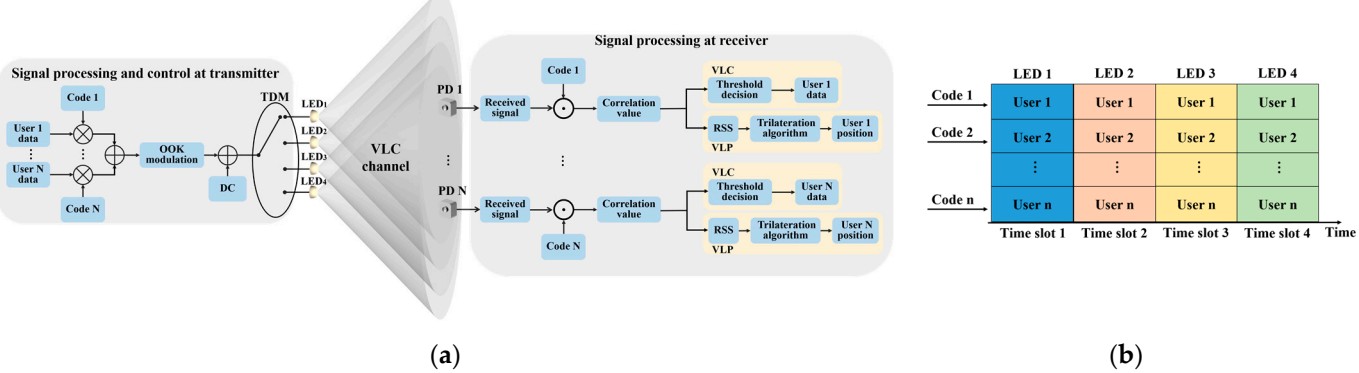

(**a**)  (**b**)

**Figure 1.** Schematic diagram of (**a**) multi-user VLCP system and (**b**) the dual-domain multiplexing scheme.

Consider there are four LEDs and $N$ users in the multi-user VLCP system and the output power without modulation of each LED lamp is $P_t$. After OOK modulation, the transmitted VLCP signal from $i$-th LED can be written as:

$$T_i(t) = P_t(\alpha S_i(t) + 1), \tag{1}$$

where $\alpha$ is the modulation index, defined as the ratio of the maximum current variation caused by the modulating signal to the bias current of LED [21,22]. The modulating signal $S_i$ is the accumulation of the encoded data of all users, which can be expressed as:

$$S_i(t) = \beta \sum_{k=1}^{N} s_k(t). \tag{2}$$

Here, we set a parameter $\beta = 1/N$ to normalize the modulating signal and $s_k$ is the encoded data of $k$-th user. The $n$-th encoded symbol of $k$-th user can be given as:

$$s_{k,n}(t) = u_{k,n}(t)c_k(t), \ (n-1)T_s \leq t < T_s, \tag{3}$$

where $u_{k,n} \in \{-1, 1\}$ is the $n$-th symbol of $k$-th user data, $c_k$ is the spreading code assigned to $k$-th user, and $T_s$ is the duration of one symbol. In this paper, we adopted the Walsh–Hadamard code as the spreading code given that it has good correlation properties [23,24]. The code waveform $c_k(t)$ of $k$-th user is expressed as:

$$c_k(t) = \sum_{l=0}^{L-1} c_{k,l} p_{T_c}(t - lT_c), \tag{4}$$

where $L$ is the length of $c_k$, $T_c$ is the duration of one chip, $c_{k,l} = c_{k,l+L} \in \{-1, 1\}$ is the $l$-th component of $c_k$, and $p_{T_c}(t)$ is a unit rectangular pulse written as:

$$p_{T_c}(t) = \begin{cases} 1 & 0 \leq t < T_c \\ 0 & \text{otherwise} \end{cases}. \tag{5}$$

Then, the VLCP signal goes through the VLC channel, which is on the basis of the Lambertian radiation model in [25], where the researchers indicated that the reflected signal is much lower than that of the light-of-sight (LOS) component. Thus, to simplify

the model, we only considered the LOS component in this paper. We also assumed that the light-emitting plane of the LED is parallel to the receiving plane of the PD. Thus, the channel DC gain from $i$-th LED to $k$-th user can be obtained from the following formula:

$$H_{i,k} = \frac{(m_l + 1)A_r g_f g_c h^{m_l+1}}{2\pi} \frac{1}{D_{i,k}^{m_l+3}}, \tag{6}$$

where $m_l$ is Lambertian order, $A_r$ is the effective receiving area of PD, $g_f$ is the gain of an optical filter, $g_c$ is the gain of an optical concentrator, and $h$ and $D_{i,k}$ are the vertical distance and Euclidean distance between $i$-th LED and $k$-th user, respectively.

We assumed that the VLCP system is perfectly synchronized. Thus, after removing the DC component, the received VLCP signal of $k$-th user can be expressed as:

$$R_{i,k}(t) = \gamma H_{i,k} P_t \alpha S_i(t) + N_{\text{AWGN}}(t), \tag{7}$$

where $\gamma$ is the PD responsivity and $N_{\text{AWGN}}$ is the additive white Gaussian noise (AWGN), including shot noise $N_{shot}$ and thermal noise $N_{thermal}$, which are described as:

$$\begin{aligned} N_{shot} &= 2q\left[\gamma P_t H_{i,k} + I_{bg}I_2\right]B \\ N_{thermal} &= 8\pi k T_k \eta A_r B^2\left(\frac{I_2}{G} + \frac{2\pi\Gamma}{g_m}\eta A_r I_3 B\right) \end{aligned}. \tag{8}$$

Here, $q$ is the electronic charge, $I_{bg}$ is the background current; $I_2$ is noise bandwidth factor; $B$ is the equivalent noise bandwidth, which is equal to the system modulation bandwidth; $k$ denotes the Boltzmann constant; $T_k$ is absolute temperature; $\eta$ is fixed capacitance of photodetector per unit area; $G$ is open loop voltage gain; $\Gamma$ is FET channel noise factor; $g_m$ is FET transconductance; and $I_3$ is gate-induced drain leakage.

For the $k$-th user, $c_k$ can be used to decode the received VLCP signal $R_{i,k}$. Then, the normalized correlation value between $n$-th encoded symbol and $c_k$ can be expressed as:

$$\begin{aligned} \xi_{i,k,n} &= \frac{1}{L}\sum_{l=0}^{L-1} R_{i,k,n}(l)c_k(l) \\ &= \gamma H_{i,k} P_t \alpha \beta \frac{1}{L}\sum_{l=0}^{L-1}\sum_{m}^{N} u_{m,n}(l)c_m(l)c_k(l) + N_{\text{noise}} \end{aligned}, \tag{9}$$

where $N_{\text{noise}} = (1/L)\sum c_k \times N_{\text{AWGN}}$ is the AWGN spread by $c_k$. Given the orthogonality of Walsh–Hadamard code, the correlation between different codes should be zero, which means:

$$\sum_{l=0}^{L-1} c_m(l)c_k(l) = \begin{cases} L & m = k \\ 0 & m \neq k \end{cases} \tag{10}$$

Therefore, Equation (9) can be rewritten as:

$$\xi_{i,k,n} = \gamma H_{i,k} P_t \alpha \beta u_{k,n} + N_{\text{noise}} \tag{11}$$

For VLC, by comparing the correlation value obtained from Equation (11) with threshold value $\lambda$, the $n$-th symbol of $k$-th user data can be recovered from:

$$v_{k,n} = \begin{cases} 1, & \xi_{i,k,n} > \lambda \\ -1, & \xi_{i,k,n} < \lambda \end{cases}. \tag{12}$$

For VLP, the RSS from $i$-th LED to $k$-th user can be calculated according to correlation value:

$$\Re_{i,k,n} = \left|\xi_{i,k,n}\right| = \gamma H_{i,k} P_t \alpha \beta + N_{\text{noise}}. \tag{13}$$

Then, by substituting Equation (6) into Equation (13), the estimated distance between *i*-th LED and *k*-th user can be obtained from:

$$\widetilde{D}_{i,k} = \varsigma(\Re_{i,k,n} - N_{\text{noise}})^{-[1/(m_l+3)]},$$
$$\text{where } \varsigma = \left( \frac{(m_l+1)A_r g_f g_c h^{m_l+1}}{2\pi} \gamma\alpha\beta P_t \right)^{[1/(m_l+3)]}. \tag{14}$$

For a specific system, the parameter $\varsigma$ is constant. After gaining the distances between all LEDs and *k*-th user, the position of *k*-th user can be located through the trilateration algorithm from [26]:

$$AE = B$$
$$\text{where } A = \begin{bmatrix} X_2 - X_1 & Y_2 - Y_1 \\ \vdots & \vdots \\ X_i - X_1 & Y_i - Y_1 \end{bmatrix}, E = \begin{bmatrix} x_k \\ y_k \end{bmatrix}, \text{ and } B = \frac{1}{2} \begin{bmatrix} X_2^2 + Y_2^2 - (X_1^2 + Y_1^2) + \widetilde{D}_{1,k}^2 - \widetilde{D}_{2,k}^2 \\ \vdots \\ X_i^2 + Y_i^2 - (X_1^2 + Y_1^2) + \widetilde{D}_{1,k}^2 - \widetilde{D}_{i,k}^2 \end{bmatrix}. \tag{15}$$

Here, $(X_i, Y_i)$ and $(x_k, y_k)$ are the coordinates of *i*-th LED and *k*-th user, respectively. Given that the measurement error of the signal transmission distance will bring obstacles to solving the above formula, the least squares estimation (LSE) method is employed to solve Equation (15) [27]. Then, the position of the receiver can be obtained by the following formula:

$$E = \left( A^T A \right)^{-1} A^T B \tag{16}$$

### 2.2. Simulation Setup

The simulation model is shown in Figure 2, where four LEDs and multiple PDs (representing multiple users) were horizontally placed on the roof and ground in a room with a size of 5 m × 5 m × 3.25 m, respectively. The X-Y coordinate of these four LEDs were $LED_1$ (−1.25 m, 1.25 m), $LED_2$ (1.25 m, 1.25 m), $LED_3$ (−1.25 m, −1.25 m), and $LED_4$ (1.25 m, −1.25 m), respectively. We set the LED light source as Lambertian radiation and ignored the reflected light. Then, 51 × 51 testing points were evenly placed in the receiving plane with a size of 5 m × 5 m, with each testing point able receive the signals from all LEDs. We evaluated the system performance under different numbers of users and various lengths of Walsh–Hadamard code. The other simulation parameters are shown in Table 1.

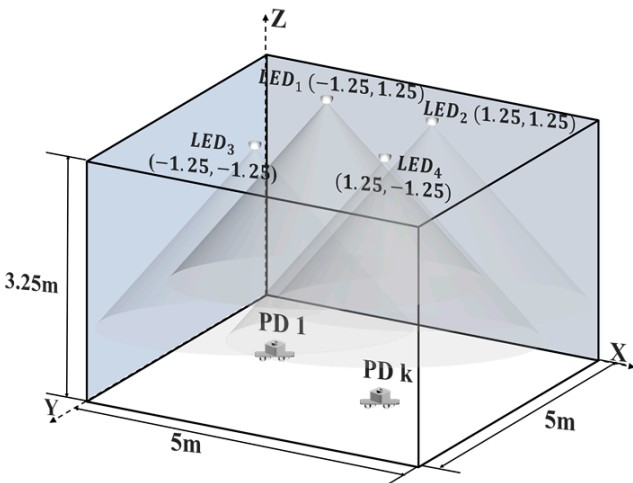

**Figure 2.** Diagram of the simulation model for the multi-user VLCP system.

**Table 1.** Simulation parameters.

| Parameters | Values |
| --- | --- |
| Output power of each LED ($P_t$) | 15 W |
| Threshold value ($\lambda$) | 0 |
| Lambertian order of emission ($m_l$) | 1 |
| Photodetector responsivity ($\gamma$) | 0.4 A/W |
| Modulation index ($\alpha$) | 0.15 |
| Physical area of Rx ($A_r$) | 75.4 mm$^2$ |
| Gain of an optical filter ($g_f$) | 1 |
| Gain of an optical concentrator ($g_c$) | 2.4115 |
| Equivalent noise bandwidth ($B$) | 10 MHz |
| Electronic charge ($q$) | $1.6 \times 10^{-19}$ C |
| Background current ($I_{bg}$) | 5100 uA |
| Noise bandwidth factor ($I_2$) | 0.562 |
| Denotes Boltzmann's constant ($k$) | $1.38064852 \times 10^{-23}$ m$^2$ kg s$^{-2}$ K$^{-1}$ |
| Absolute temperature ($T_k$) | 295 K |
| Fixed capacitance of photodetector per unit area ($\eta$) | 112 pF/cm$^2$ |
| Open loop voltage gain ($G$) | 10 |
| FET channel noise factor ($\Gamma$) | 1.5 |
| FET transconductance ($g_m$) | 30 mS |
| Gate induced drain leakage ($I_3$) | 0.0868 |

*2.3. Experiment Setup*

We also established an experimental platform to validate the proposed multi-user VLCP system based on the dual-domain multiplexing scheme, as shown in Figure 3. Four commercial LED lamps and one PD (THORLABS PDA100A2) were placed on the top and bottom of the cuboid frame, respectively. The output power of each LED was 5 W, and the photodetector responsivity and physical area of PD were the same as the simulation setup. The X-Y coordinates of these four LEDs were $LED_1$ (25 cm, 25 cm), $LED_2$ (−25 cm, 25 cm), $LED_3$ (25 cm, −25 cm) and $LED_4$ (−25 cm, 2–5 cm), respectively, and the vertical distance between LEDs and PD was 90 cm. We assumed that there were 8 users and each user was assigned a unique Walsh–Hadamard code, with a length of 32 as spreading code. Firstly, we randomly generated 8 sets of user data and each frame included 100 bits. These user data were encoded by the corresponding spreading code and then accumulated based on Equation (2) to generate the transmission data. Then, we used two arbitrary waveform generators (AWG RIGOL DG2102) to modulate the transmission data and load the signal to four LEDs in turn. Here, the peak-to-peak voltage, bias voltage, and sampling rate were set as 1.05 V, 3.50 V, and 100 KSa/s, respectively. Limited by the equipment, we only used one PD to receive the VLCP signals and used one oscilloscope (OSC RIGOL MSO5354) to collect the received signals at the sampling rate of 1 MSa/s. Finally, we copied the data to the computer and successively decoded it by using the spreading code of each user to realize the VLC and VLP functions. Given that the environment of an experiment is more complex than that of a simulation, we adopted a fitting method to correct the parameters in Equation (14) and calculate different transmission distances [28]. To evaluate the system performance, we moved the PD onto the receiving plane, with a size of 60 cm × 60 cm in the middle of the bottom of the cuboid frame, where 12 × 12 testing points were evenly distributed on the receiving plane. The experimental results will be discussed later.

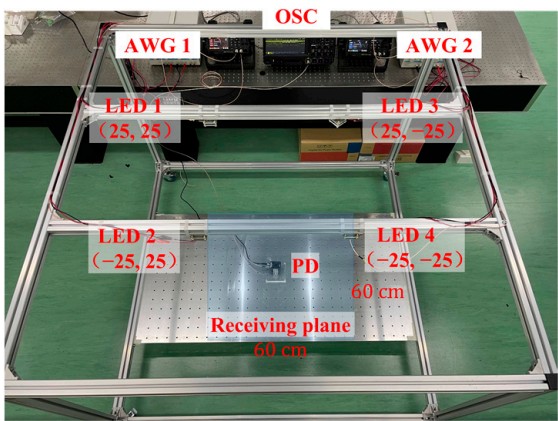

**Figure 3.** Diagram of the experimental platform for the multi-user VLCP system.

## 3. Results and Discussion

### 3.1. Simulation Results and Discussion

Firstly, we evaluated the communication and positioning performances of the multi-user VLCP system under different numbers of users. Here the length (L) of Walsh-Hadamard code was fixed at 32 and the number (U) of users was set as 8, 12, and 16, respectively. We estimated the BER and positioning error at each testing point, where the positioning error was defined as the Euclidean distance between real position and estimated position. Figures 4 and 5 show the surface diagram of BER and positioning error, respectively. We can see that BER and positioning error estimated in the center area of the receiving plane were much lower than in the corner area. This is because the PD in the center area can receive a larger optical power from all LEDs compared to the PD in the corner area. In addition, both BER and positioning error rose rapidly with the increase of user numbers because all the users equally shared the output power of LEDs, according to Equation (2). When the user number increased, the power assigned to each user decreased, leading to a lower signal to noise ratio (SNR). Even so, most testing points had BER within a forward error correction (FEC) citation of $3.8 \times 10^{-3}$ and positioning error less than 10 cm. We calculated the cumulative distribution function (CDF) value of BER and positioning error estimated in the whole receiving plane. Figure 6 shows that more than 96%, 85%, and 74% of testing points had the BER within FEC citation and more than 85%, 72%, and 59% of testing points had positioning errors less than 5 cm when the user number was 8, 12, and 16, respectively.

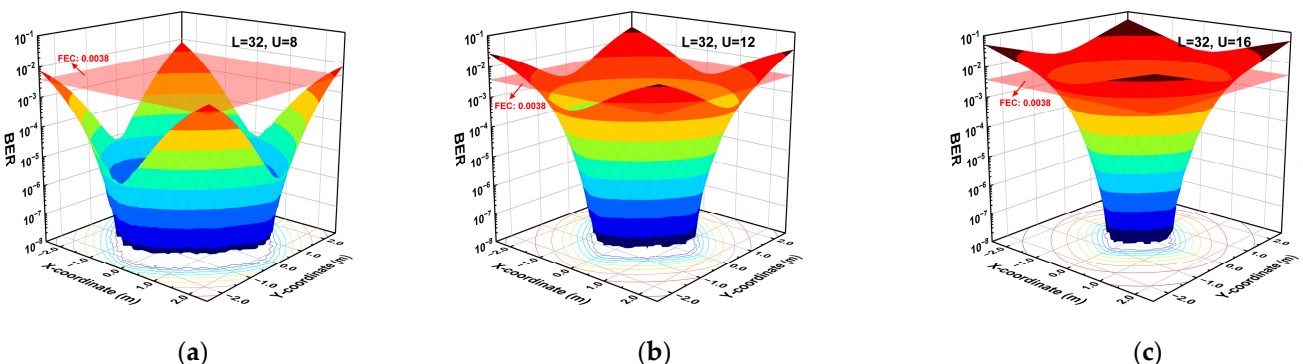

(**a**)                           (**b**)                           (**c**)

**Figure 4.** Surface diagram of BER estimated at all testing points when the user number is (**a**) 8 users, (**b**) 12 users, and (**c**) 16 users.

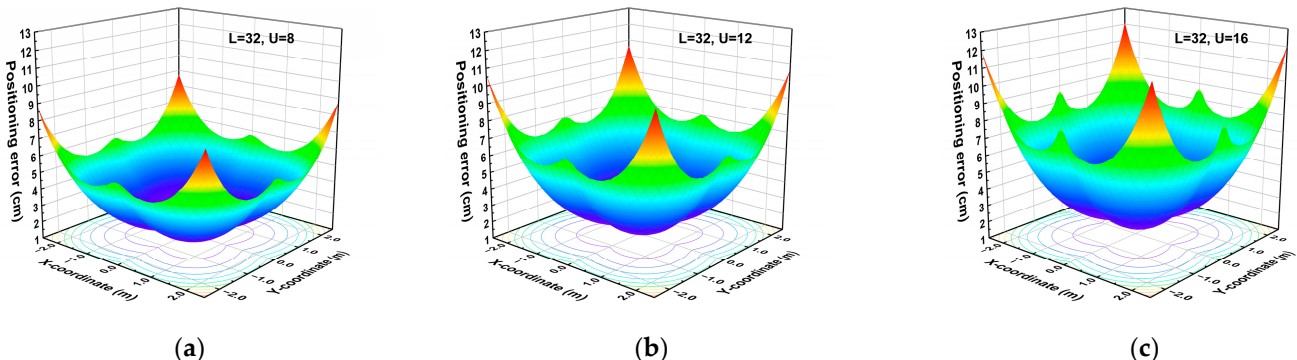

**Figure 5.** Surface diagram of positioning error estimated at all testing points when the user number is (**a**) 8 users, (**b**) 12 users, and (**c**) 16 users.

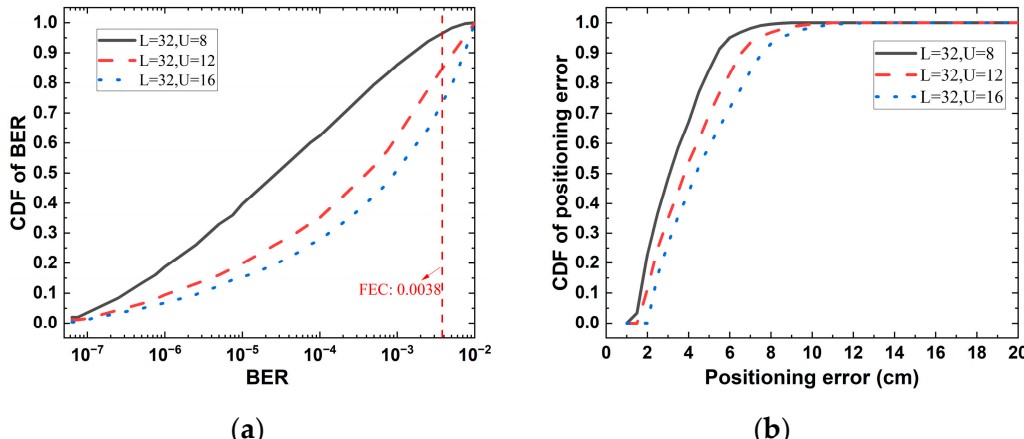

**Figure 6.** CDF of (**a**) BER and (**b**) positioning error estimated in the whole receiving plane when the multi-user VLCP system supports different numbers of users.

Then, we investigated the performance of the eight-user VLCP system when adopting the Walsh–Hadamard code with lengths of 16, 32, and 64, respectively. The surface diagram of BER and positioning error are displayed in Figures 7 and 8, respectively, and the CDF of BER and positioning error are shown in Figure 9. The figures show that the measured BER of 87%, 96%, and 100% of testing points were lower than $3.8 \times 10^{-3}$ and the measured positionings of 59%, 85%, and 95% of testing points were less than 5 cm when the length of the spreading code was 16, 32, and 64, respectively. In addition, we found that the BER and positioning error can be significantly reduced by using a Walsh–Hadamard code with a longer length. This is because the AWGN is spread by the Walsh–Hadamard code according to Equation (9) when we decoded the signal. Thus, the code with a longer length can equalize much AWGN, leading to higher SNR, and consequently, lower BER and positioning error. Additionally, with the increased length of Walsh–Hadamard code, the number of spreading codes in one set can be increased. Thus, the system can support more users simultaneously. However the longer length of spreading code will lower the data rate when the sampling rate of transmission signal is fixed. Therefore, when designing the multi-user VLCP system, it is necessary to choose a suitable length of spreading code according to the required supporting user number, data rate, and positioning accuracy.

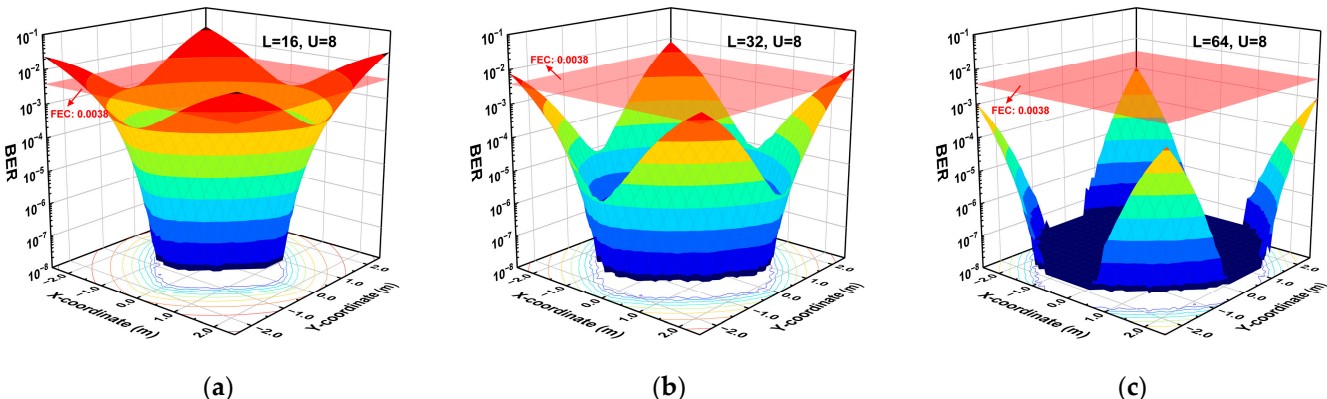

**Figure 7.** Surface diagram of BER estimated at all testing points of the 8-user VLCP system when the length of spreading code is (**a**)16, (**b**) 32, and (**c**) 64.

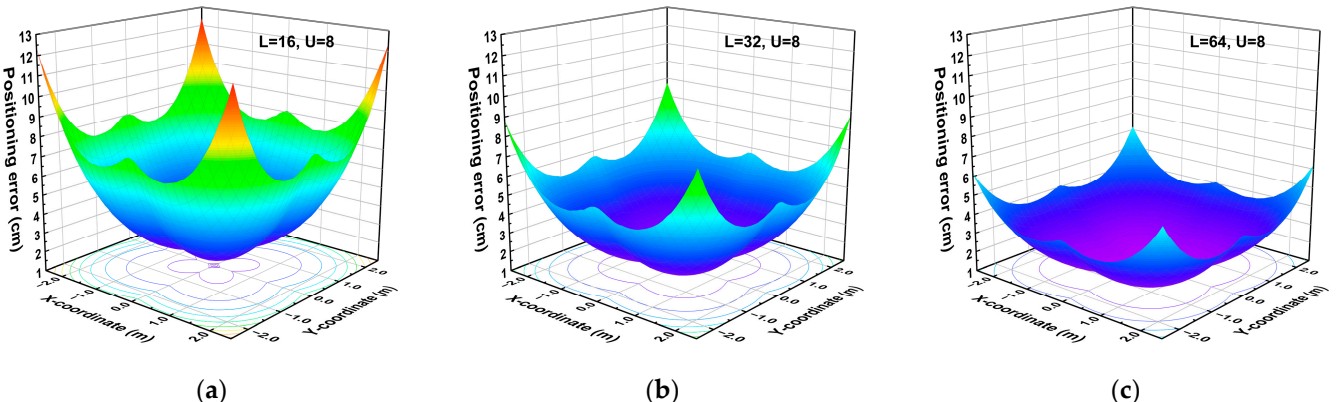

**Figure 8.** Surface diagram of positioning error estimated at all testing points of the 8-user VLCP system when the length of spreading code is (**a**)16, (**b**) 32, and (**c**) 64.

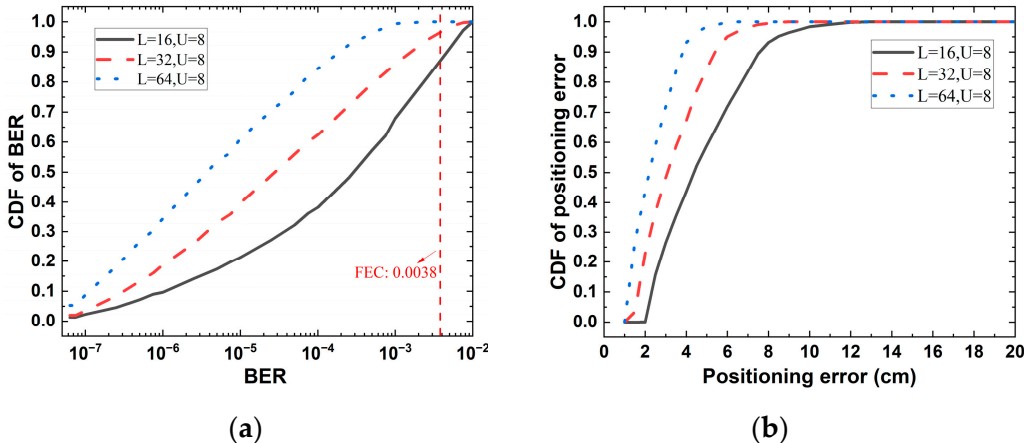

**Figure 9.** CDF of (**a**) BER and (**b**) positioning error estimated in the whole receiving plane when the 8-user VLCP system adopts spreading codes with different length.

### 3.2. Experimental Results and Discussion

Firstly, we evaluated the communication performance at three specific testing points, including points A (25 cm, 25 cm), B (0 cm, 0 cm), and C (−30 cm, −30 cm), where each testing point received $1e^7$ frames of data ($1e^7 \times 100 = 1e^9$ bits). Frames of received waveforms of four LEDs after sampling are shown in Figure 10. We can see that the PD received clear waveforms from each LED at these specific testing points. Based on the

received waveforms, all of the user data were able to be entirely recovered by using the spreading codes according to Equation (12). Therefore, the PD at these specific testing points was able to achieve error-free communication.

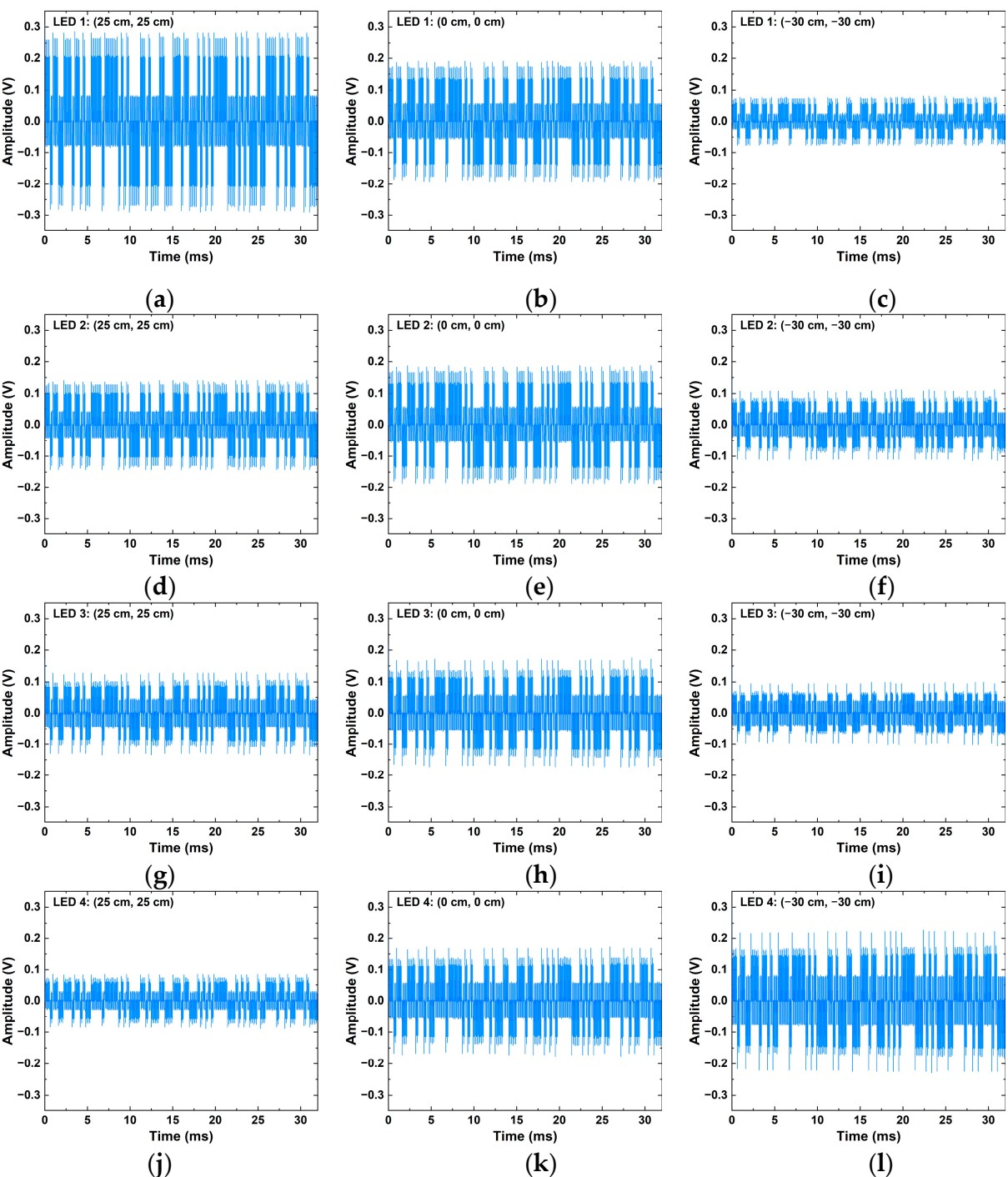

**Figure 10.** Frames of received waveforms of LED 1 (**a**) measured at point A, (**b**) measured at point B, and (**c**) measured at point C after sampling; frames of received waveforms of LED 2 (**d**) measured at point A, (**e**) measured at point B, and (**f**) measured at point C after sampling; frames of received waveforms of LED 3 (**g**) measured at point A, (**h**) measured at point B, and (**i**) measured at point C after sampling; and frames of received waveforms of LED 4 (**j**) measured at point A, (**k**) measured at point B, and (**l**) measured at point C after sampling.

For the positioning performance, we estimated the RSS of each LED at all testing points first. The RSS distributions of four LEDs showed a similar and clear trend, as shown in Figure 11, where the RSS became larger when the PD was closer to the LED. Based on the estimated RSS of each LED, we adopted the fitting method to correct Equation (14). Thus, transmission distance between four LEDs and the PD was able to be calculated from:

$$
\begin{cases}
\widetilde{D}_1 = 40.42(\Re_1)^{-0.2278} + 1.47 \\
\widetilde{D}_2 = 30.21(\Re_2)^{-0.2658} + 14.06 \\
\widetilde{D}_3 = 20.24(\Re_3)^{-0.3217} + 29.08 \\
\widetilde{D}_4 = 19.15(\Re_4)^{-0.3334} + 29.81
\end{cases}.
\tag{17}
$$

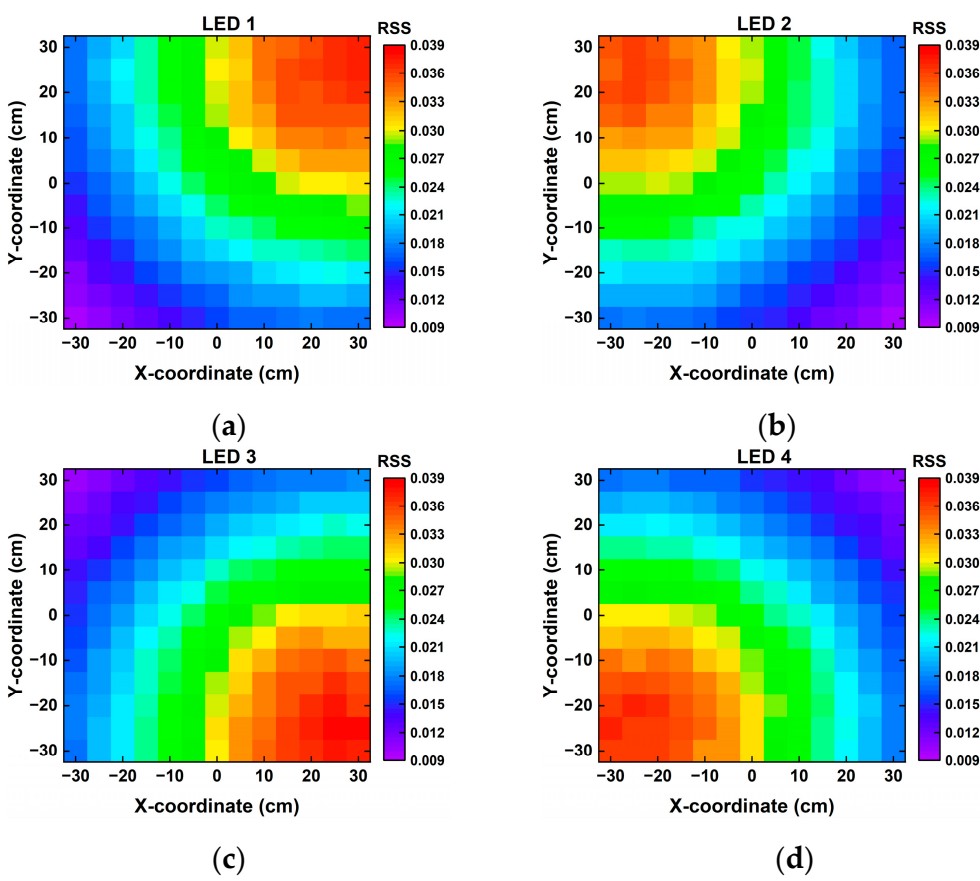

**Figure 11.** The RSS distributions of (**a**) LED 1, (**b**) LED 2, (**c**) LED 3, and (**d**) LED 4.

Then, we defined the absolute value of the difference between estimated transmission distance and real distance as the error of transmission distance (ETD). The ETD distributions of four LEDs are shown in Figure 12, where the closer PD was to the LED, the lower ETD. This is because the area near the LED had the better SNR. Finally, with the measured transmission, we estimated the PD location by using Equation (16). Figure 13 displays the positioning results. We found that the center area of the receiving plane had better positioning performance than the corner area, which was caused by uneven illumination. From Figure 13b, there was an approximate 96% positioning error under 2.0 cm, and the average positioning error was 1.08 cm. Therefore, the proposed VLCP system was able to realize an accurate positioning with a 2 cm positioning accuracy.

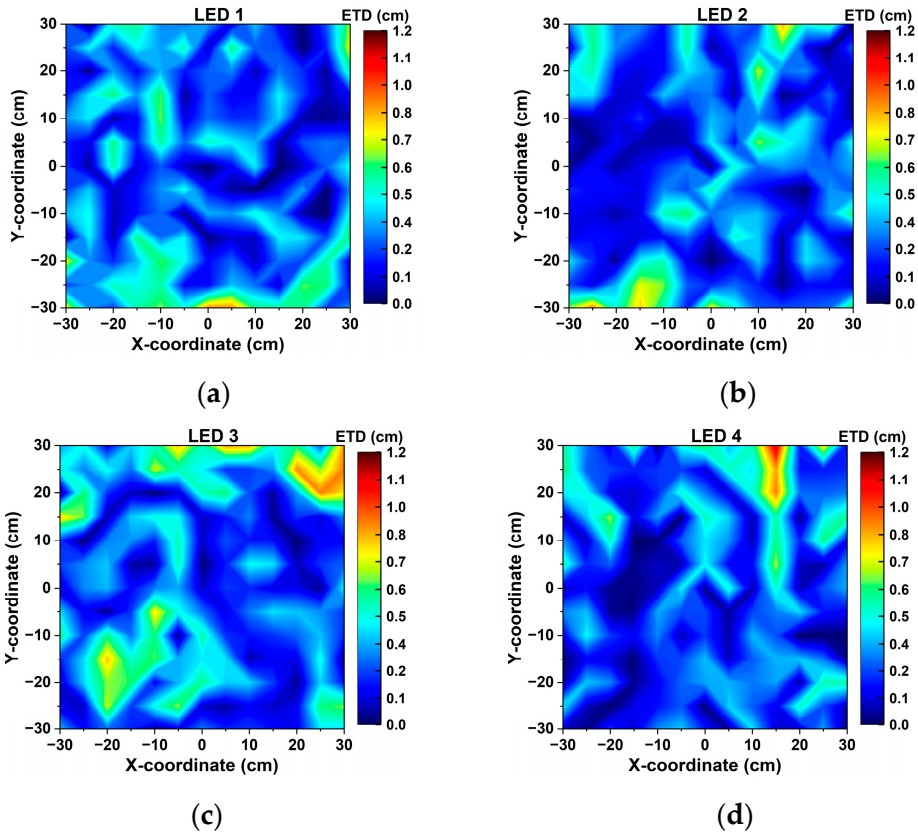

**Figure 12.** Distribution ETD of (**a**) LED 1, (**b**) LED 2, (**c**) LED 3, and (**d**) LED 4.

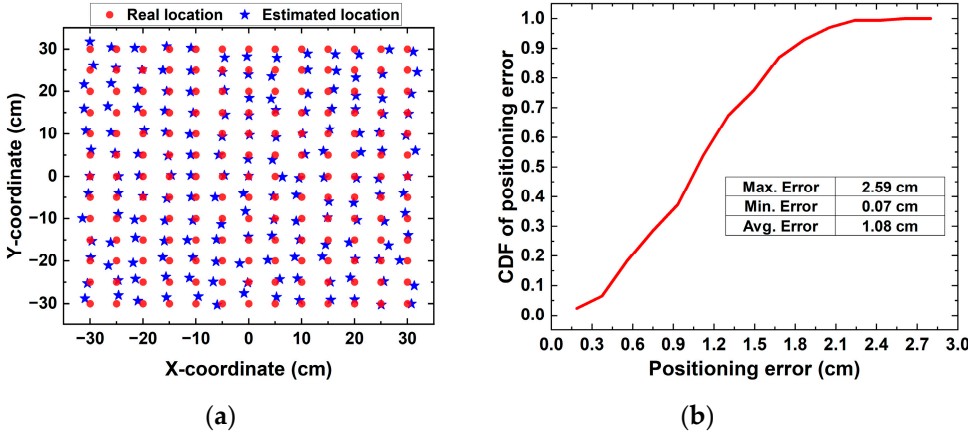

**Figure 13.** (**a**)The real location and estimated location of each testing point and (**b**) the CDF of positioning error.

## 4. Conclusions

A multi-user VLCP system based on the dual-domain multiplexing scheme is proposed, where the data of different users and the VLCP signals from different LEDs are separated through code-division multiplexing and time-division multiplexing, respectively. A set of Walsh–Hadamard codes is assigned to multiple users, and its orthogonality is utilized to distinguish different users. The correlation values of the decoding results are used to recover the user data for VLC and calculate the RSS for VLP. Thus, the proposed method can effectively integrate VLC and VLP into one system and simultaneously support multiple users. The supporting user number is not limited by the number of LEDs but can be increased by adopting the Walsh–Hadamard code with a longer length. Both

the simulation and experimental testing were investigated to verify the feasibility of the proposed VLCP system.

**Author Contributions:** Conceptualization, Z.L.; methodology, Z.L.; software, Z.L.; validation, Z.L.; formal analysis, Z.L.; investigation, Z.L.; resources, Z.L.; data curation, Z.L.; writing—original draft preparation, Z.L.; writing—review and editing, Z.L.; visualization, Z.L.; supervision, C.Y.; project administration, C.Y.; funding acquisition, C.Y. All authors have read and agreed to the published version of the manuscript.

**Funding:** This research was funded by Hong Kong Research Grants Council, grant number GRF15212720, and Key Basic Research Scheme of Shenzhen Natural Science Foundation, grant number JCYJ20200109142010888.

**Institutional Review Board Statement:** Not applicable.

**Informed Consent Statement:** Not applicable.

**Data Availability Statement:** Data underlying the results presented in this paper are not publicly available at this time but may be obtained from the authors upon reasonable request.

**Conflicts of Interest:** The authors declare no conflict of interest.

## Nomenclature

| Abbreviation | Description |
| --- | --- |
| VLC | Visible light communication |
| VLP | Visible light positioning |
| VLCP | visible light communication and positioning |
| 6G | sixth-generation |
| IoT | Internet of Things |
| RF | radio frequency |
| TDM | time-division multiplexing |
| FDM | frequency-division multiplexing |
| CDM | code-division multiplexing |
| DDM | dual-domain multiplexing |
| RSS | receiver signal strength |
| PD | photodetector |
| OOK | on-off keying |
| LOS | light-of-sight |
| AWGN | additive white Gaussian noise |
| LSE | least squares estimation |
| SNR | signal to noise ratio |
| BER | bit-error rate |
| FEC | forward error correction |
| CDF | cumulative distribution function |
| ETD | error of transmission distance |

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
