# Peer review of "Multi-User Visible Light Communication and Positioning System Based on Dual-Domain Multiplexing Scheme"

_photonics, doi:10.3390/photonics10030306_

Round 1

Reviewer 1 Report

This paper proposed a dual-domain multiplexing scheme for a multi-user VLCP system both with simulation and experiment. Some minor issues are given as follows:

1. The authors said 'Walsh-Hadamard code has perfect orthogonality', which is not rigorous. ‘Good correlation properties’ should be better.

2. In line 158, the authors said 'by substituting Eq.(6) into Eq.(13)', where is Nnoise? About 'noise', please explain.

3. The authors need to give more details about the experimental system, such as the transmitted power of LEDs.

Reviewer 2 Report

The work is interesting, but even the authors should better justify the importance, advantages, and disadvantages of other positioning systems. Why it is better against RF systems is not clear. A background or state-of-the-art section is needed.

I recommend putting a section of acronyms that helps to identify all the terms.

Reviewer 3 Report

The paper presents a VLC and VLP system based on time division and code division multiplexing technology for multiple users simultaneously.

An important part of the contribution is the evaluation through simulation and experimentally.

The manuscript is well-written in general, and the discussions are interesting. I am happy to recommend publication once the following points are addressed

a.     There is no discusión why White LED is used instead RGB LED. How would the results change if an RGB LED is used?

b.     The authors mention that the contribution of NLOS is much smaller than LOS (line 124), please justify it.

The NLOS contribution depends on several factors, such as the size of the stage, material of the walls, floor and ceiling as well as the position of the receiver. Review the model proposed in [1] for eq (6).

[1] Zhang, X.; Duan, J.; Fu, Y.; Shi, A. Theoretical accuracy analysis of indoor visible light communication positioning system based on Received Signal Strength Indicator. J. Lightw. Technol. 2014, 32, 4180–4186.

c. For a more realistic system, what changes would you expect to see?

  d. The comparison of the simulated and experimental results is in doubt due to the difference in the size of the stage.

Round 2

Reviewer 2 Report

Thank you, the Authors, for addressing all my comments.